# GENERATIVE CODE MODELING WITH GRAPHS

**Marc Brockschmidt, Miltiadis Allamanis, Alexander Gaunt**
Microsoft Research
Cambridge, UK
{mabrocks,miallama,algaunt}@microsoft.com

**Oleksandr Polozov**
Microsoft Research
Redmond, WA, USA
polozov@microsoft.com

## ABSTRACT

Generative models for source code are an interesting structured prediction problem, requiring to reason about both hard syntactic and semantic constraints as well as about natural, likely programs. We present a novel model for this problem that uses a graph to represent the intermediate state of the generated output. Our model generates code by interleaving grammar-driven expansion steps with graph augmentation and neural message passing steps. An experimental evaluation shows that our new model can generate semantically meaningful expressions, outperforming a range of strong baselines.

## 1 INTRODUCTION

Learning to understand and generate programs is an important building block for procedural artificial intelligence and more intelligent software engineering tools. It is also an interesting task in the research of structured prediction methods: while imbued with formal semantics and strict syntactic rules, *natural* source code carries aspects of natural languages, since it acts as a means of communicating intent among developers. Early works in the area have shown that approaches from natural language processing can be applied successfully to source code (Hindle et al., 2012), whereas the programming languages community has had successes in focusing exclusively on formal semantics. More recently, methods handling both modalities (*i.e.*, the formal and natural language aspects) have shown successes on important software engineering tasks (Raychev et al., 2015; Bichsel et al., 2016; Allamanis et al., 2018b) and semantic parsing (Yin & Neubig, 2017; Rabinovich et al., 2017).

However, current *generative* models of source code mostly focus on only one of these modalities at a time. For example, program synthesis tools based on enumeration and deduction (Solar-Lezama, 2008; Polozov & Gulwani, 2015; Feser et al., 2015; Feng et al., 2018) are successful at generating programs that satisfy some (usually incomplete) formal specification but are often obviously wrong on manual inspection, as they cannot distinguish unlikely from likely, "natural" programs. On the other hand, learned code models have succeeded in generating realistic-looking programs (Maddison & Tarlow, 2014; Bielik et al., 2016; Parisotto et al., 2017; Rabinovich et al., 2017; Yin & Neubig, 2017). However, these programs often fail to be semantically relevant, for example because variables are not used consistently.

In this work, we try to overcome these challenges for generative code models and present a general method for generative models that can incorporate structured information that is deterministically available at generation time. We focus our attention on generating source code and follow the ideas of *program graphs* (Allamanis et al., 2018b) that have been shown to learn semantically meaningful representations of (pre-existing) programs. To achieve this, we lift grammar-based tree decoder models into the graph setting, where the diverse relationships between various elements of the generated code can be modeled. For this, the syntax tree under generation is augmented with additional edges denoting known relationships (*e.g.*, last use of variables). We then interleave the steps of the generative procedure with neural message passing (Gilmer et al., 2017) to compute more precise representations of the intermediate states of the program generation. This is fundamentally different from sequential generative models of graphs (Li et al., 2018; Samanta et al., 2018), which aim to generate all edges and nodes, whereas our graphs are deterministic augmentations of generated trees.

To summarize, we present *a)* a general graph-based generative procedure for highly structured objects, incorporating rich structural information; *b)* ExprGen, a new code generation task focused on

**Algorithm 1** Pseudocode for Expand

---

**Input:** Context $c$, partial AST $a$, node $v$ to expand
1: $\mathbf{h}_v \leftarrow$ getRepresentation$(c, a, v)$
2: $rhs \leftarrow$ pickProduction$(v, \mathbf{h}_v)$
3: **for** child node type $\ell \in rhs$ **do**
4: $\quad (a, u) \leftarrow$ insertChild$(a, \ell)$
5: $\quad$ **if** $\ell$ is nonterminal type **then**
6: $\quad\quad a \leftarrow$ Expand$(c, a, u)$
7: **return** $a$

---

```
int ilOffsetIdx =
  Array.IndexOf(sortedILOffsets, map.ILOffset);
int nextILOffsetIdx = ilOffsetIdx + 1;
int nextMapILOffset =
  nextILOffsetIdx < sortedILOffsets.Length
  ? sortedILOffsets[nextILOffsetIdx]
  : int.MaxValue;
```

Figure 1: Example for ExprGen, target expression to be generated is `marked`. Taken from `BenchmarkDotNet`, lightly edited for formatting.

generating small, but semantically complex expressions conditioned on source code context; and *c)* a comprehensive experimental evaluation of our generative procedure and a range of baseline methods from the literature.

## 2 BACKGROUND & TASK

The most general form of the code generation task is to produce a (partial) program in a programming language given some context information $c$. This context information can be natural language (as in, *e.g.*, semantic parsing), input-output examples (*e.g.*, inductive program synthesis), partial program sketches, *etc.* Early methods generate source code as a sequence of tokens (Hindle et al., 2012; Hellendoorn & Devanbu, 2017) and sometimes fail to produce syntactically correct code. More recent models are sidestepping this issue by using the target language's grammar to generate abstract syntax trees (ASTs) (Maddison & Tarlow, 2014; Bielik et al., 2016; Parisotto et al., 2017; Yin & Neubig, 2017; Rabinovich et al., 2017), which are syntactically correct by construction.

In this work, we follow the AST generation approach. The key idea is to construct the AST $a$ sequentially, by expanding one node at a time using production rules from the underlying programming language grammar. This simplifies the code generation task to a sequence of classification problems, in which an appropriate production rule has to be chosen based on the context information and the partial AST generated so far. In this work, we simplify the problem further — similar to Maddison & Tarlow (2014); Bielik et al. (2016) — by fixing the order of the sequence to always expand the left-most, bottom-most nonterminal node. Alg. 1 illustrates the common structure of AST-generating models. Then, the probability of generating a given AST $a$ given some context $c$ is

$$p(a \mid c) = \prod_t p(a_t \mid c, a_{<t}), \tag{1}$$

where $a_t$ is the production choice at step $t$ and $a_{<t}$ the partial syntax tree generated before step $t$.

**Code Generation as Hole Completion** We introduce the ExprGen task of filling in code within a hole of an otherwise existing program. This is similar, but not identical to the auto-completion function in a code editor, as we assume information about the following code as well and aim to generate whole expressions rather than single tokens. The ExprGen task also resembles program sketching (Solar-Lezama, 2008) but we give no other (formal) specification other than the surrounding code. Concretely, we restrict ourselves to expressions that have Boolean, arithmetic or string type, or arrays of such types, excluding expressions of other types or expressions that use project-specific APIs. An example is shown in Fig. 1. We picked this subset because it already has rich semantics that can require reasoning about the interplay of different variables, while it still only relies on few operators and does not require to solve the problem of open vocabularies of full programs, where an unbounded number of methods would need to be considered.

In our setting, the context $c$ is the pre-existing code around a hole for which we want to generate an expression. This also includes the set of variables $v_1, \ldots, v_\ell$ that are in scope at this point, which can be used to guide the decoding procedure (Maddison & Tarlow, 2014). Note, however, that our method is *not* restricted to code generation and can be easily extended to all other tasks and domains that can be captured by variations of Alg. 1 (*e.g.* in NLP).

## 3 GRAPH DECODING FOR SOURCE CODE

To tackle the code generation task presented in the previous section, we have to make two design choices: (a) we need to find a way to encode the code context $c, v_1, \ldots, v_\ell$ and (b) we need to construct a model that can learn $p(a_t \mid c, a_{<t})$ well. We do *not* investigate the question of encoding the context in this paper, and use two existing methods in our experiments in Sect. 5. Both these encoders yield a distributed vector representation for the overall context, representations $\boldsymbol{h}_{t_1}, \ldots, \boldsymbol{h}_{t_T}$ for all tokens in the context, and separate representations for each of the in-scope variables $v_1, \ldots, v_\ell$, summarizing how each variable is used in the context. This information can then be used in the generation process, which is the main contribution of our work and is described in this section.

**Overview** Our decoder model follows the grammar-driven AST generation strategy of prior work as shown in Alg. 1. The core difference is in how we compute the representation of the node to expand. Maddison & Tarlow (2014) construct it entirely from the representation of its parent in the AST using a log-bilinear model. Rabinovich et al. (2017) construct the representation of a node using the parents of the AST node but also found it helpful to take the relationship to the parent node (*e.g.* "condition of a `while`") into account. Yin & Neubig (2017) on the other hand propose to take the last expansion step into account, which may have finished a subtree "to the left". In practice, these additional relationships are usually encoded by using gated recurrent units with varying input sizes.

We propose to generalize and unify these ideas using a graph to structure the flow of information in the model. Concretely, we use a variation of attribute grammars (Knuth, 1967) from compiler theory to derive the structure of this graph. We associate each node in the AST with two fresh nodes representing *inherited* resp. *synthesized* information (or attributes). Inherited information is derived from the context and parts of the AST that are already generated, whereas synthesized information can be viewed as a "summary" of a subtree. In classical compiler theory, inherited attributes usually contain information such as declared variables and their types (to allow the compiler to check that only declared variables are used), whereas synthesized attributes carry information about a subtree "to the right" (*e.g.*, which variables have been declared). Traditionally, to implement this, the language grammar has to be extended with explicit rules for deriving and synthesizing attributes.

To transfer this idea to the deep learning domain, we represent attributes by distributed vector representations and train neural networks to learn how to compute attributes. Our method for getRepresentation from Alg. 1 thus factors into two parts: a deterministic procedure that turns a partial AST $a_{<t}$ into a graph by adding additional edges that encode attribute relationships, and a graph neural network that learns from this graph.

**Notation** Formally, we represent programs as graphs where nodes $u, v, \ldots$ are either the AST nodes or their associated attribute nodes, and *typed* directed edges $\langle u, \tau, v \rangle \in \mathcal{E}$ connect the nodes according to the flow of information in the model. The edge types $\tau$ represent different syntactic or semantic relations in the information flow, discussed in detail below. We write $\mathcal{E}_v$ for the set of incoming edges into $v$. We also use functions like $\mathsf{parent}(a, v)$ and $\mathsf{lastSibling}(a, v)$ that look up and return nodes from the AST $a$ (*e.g.* resp. the parent node of $v$ or the preceding AST sibling of $v$).

**Example** Consider the AST of the expression `i - j` shown in Fig. 2 (annotated with attribute relationships) constructed step by step by our model. The AST derivation using the programming language grammar is indicated by shaded backgrounds, nonterminal nodes are shown as rounded rectangles, and terminal nodes are shown as rectangles. We additionally show the variables given within the context as dashed rectangles at the bottom. First, the root node, Expr, was expanded using the production rule $(1) : \texttt{Expr} \implies \texttt{Expr} - \texttt{Expr}$. Then, its two nonterminal children were in turn expanded to the set of known variables using the produc-

---

**Algorithm 2** Pseudocode for ComputeEdge

**Input:** Partial AST $a$, node $v$
1: Edge set $\mathcal{E} \leftarrow \varnothing$
2: **if** $v$ is inherited **then**
3:     $\mathcal{E} \leftarrow \mathcal{E} \cup \{\langle \mathsf{parent}(a, v), Child, v \rangle\}$
4:     **if** $v$ is terminal node **then**
5:         $\mathcal{E} \leftarrow \mathcal{E} \cup \{\langle \mathsf{lastToken}(a, v), NextToken, v \rangle\}$
6:         **if** $v$ is variable **then**
7:             $\mathcal{E} \leftarrow \mathcal{E} \cup \{\langle \mathsf{lastUse}(a, v), NextUse, v \rangle\}$
8:     **if** $v$ is not first child **then**
9:         $\mathcal{E} \leftarrow \mathcal{E} \cup \{\langle \mathsf{lastSibling}(a, v), NextSib, v \rangle\}$
10: **else**
11:     $\mathcal{E} \leftarrow \mathcal{E} \cup \{\langle u, Parent, v \rangle \mid u \in \mathsf{children}(a, v)\}$
12:     $\mathcal{E} \leftarrow \mathcal{E} \cup \{\langle \mathsf{inheritedAttr}(v), InhToSyn, v \rangle\}$
13: **return** $\mathcal{E}$

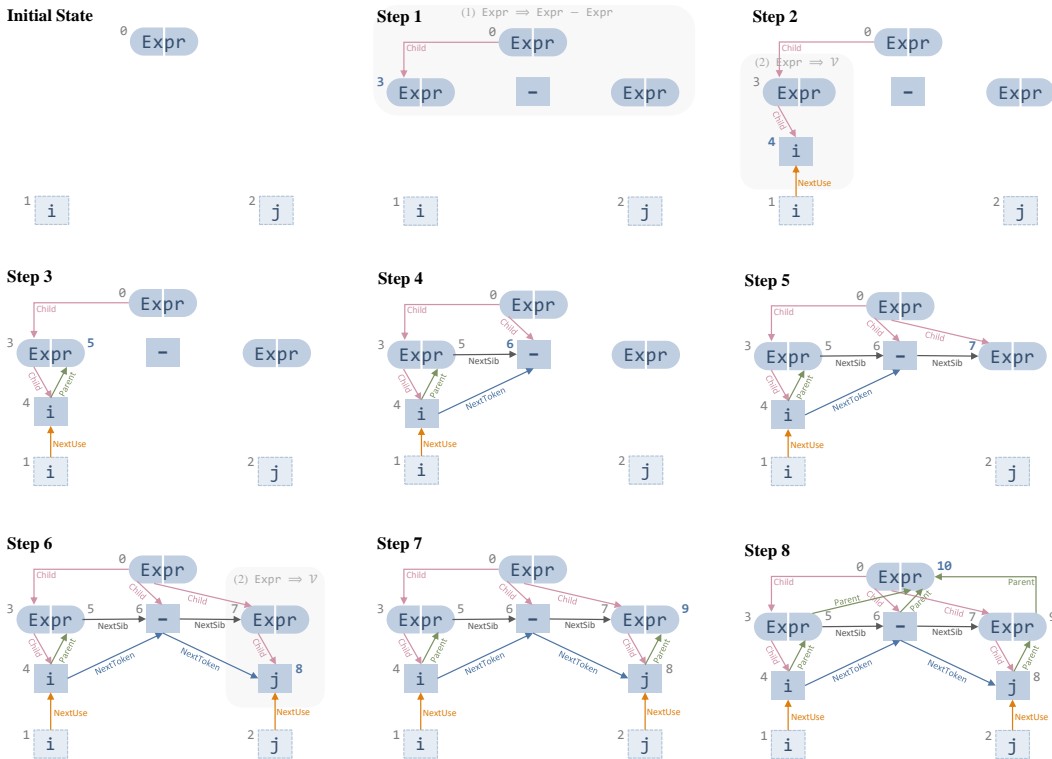

Figure 2: Example AST with attribute dependencies, shown constructed step by step in the order of generation. Each AST node (labeled by a terminal or non-terminal) has either one or two associated attribute nodes, shown as its left/right parts. The node IDs are highlighted at the corresponding generation step. Edge color and label indicate edge type. Edges are computed using Alg. 2, but are only depicted after use in message passing. Best viewed in color.

tion rule $(2)$ : $\texttt{Expr} \implies \mathcal{V}$, choosing $\texttt{i}$ for the first variable and $\texttt{j}$ for the second variable (*cf.* below for details on picking variables).

Attribute nodes are shown overlaying their corresponding AST nodes. For example, the root node is associated with its inherited attributes node $0$ and with node $10$ for its synthesized attributes. For simplicity, we use the same representation for inherited and synthesized attributes of terminal nodes.

**Edges in $a_{<t}$** We discuss the edges used in our *neural attribute grammars* ($\mathcal{NAG}$) on our example below, and show them in Fig. 2 using different edge drawing styles for different edge types. Once a node is generated, the edges connecting this node can be deterministically added to $a_{<t}$ (precisely defined in Alg. 2). The list of different edge types used in our model is as follows:

- *Child* (red) edges connect an inherited attribute node to the inherited attributes nodes of its children, as seen in the edges from node $0$. These are the connections in standard syntax-driven decoders (Maddison & Tarlow, 2014; Parisotto et al., 2017; Yin & Neubig, 2017; Rabinovich et al., 2017).

- *Parent* (green) edges connect a synthesized attribute node to the synthesized attribute node of its AST parent, as seen in the edges leading to node $10$. These are the additional connections used by the R3NN decoder introduced by Parisotto et al. (2017).

- *NextSib* (black) edges connect the synthesized attribute node to the inherited attribute node of its next sibling (*e.g.* from node $5$ to node $6$). These allow information about the synthesized attribute nodes from a fully generated subtree to flow to the next subtree.

- *NextUse* (orange) edges connect the attribute nodes of a variable (since variables are always terminal nodes, we do not distinguish inherited from synthesized attributes) to their next use. Unlike Allamanis et al. (2018b), we do not perform a dataflow analysis, but instead just

follow the lexical order. This can create edges from nodes of variables in the context $c$ (for example, from node 1 to 4 in Fig. 2), or can connect AST leaf nodes that represent multiple uses of the same variable within the generated expressions.

- *NextToken* (blue) edges connect a terminal node (a token) to the next token in the program text, for example between nodes 4 and 6.
- *InhToSyn* edges (not shown in Fig. 2) connect the inherited attributes nodes to its synthesized attribute nodes. This is not strictly adding any information, but we found it to help with training.

The panels of Fig. 2 show the timesteps at which the representations of particular attribute nodes are computed and added to the graph. For example, in the second step, the attributes for the terminal token `i` (node 4) in Fig. 2 are computed from the inherited attributes of its AST parent `Expr` (node 3), the attributes of the last use of the variable `i` (node 1), and the node label `i`. In the third step, this computed attribute is used to compute the synthesized attributes of its AST parent `Expr` (node 5).

**Attribute Node Representations**    To compute the neural attribute representation $\mathbf{h}_v$ of an attribute node $v$ whose corresponding AST node is labeled with $\ell_v$, we first obtain its incoming edges using Alg. 2 and then use the state update function from Gated Graph Neural Networks (GGNN) (Li et al., 2016). Thus, we take the attribute representations $\mathbf{h}_{u_i}$ at edge sources $u_i$, transform them according to the corresponding edge type $t_i$ using a learned function $f_{t_i}$, aggregate them (by elementwise summation) and combine them with the learned embedding $\mathsf{emb}(\ell_v)$ of the node label $\ell_v$ using a function $g$:

$$\mathbf{h}_v = g(\mathsf{emb}(\ell_v), \sum_{\langle u_i, t_i, v \rangle \in \mathcal{E}_v} f_{t_i}(\mathbf{h}_{u_i})) \tag{2}$$

In practice, we use a single linear layer for $f_{t_i}$ and implement $g$ as a gated recurrent unit (Cho et al., 2014). We compute node representations in such an order that all $\mathbf{h}_{u_i}$ appearing on the right of (2) are already computed. This is possible as the graphs obtained by repeated application of Alg. 2 are directed acyclic graphs rooted in the inherited attribute node of the root node of the AST. We initialize the representation of the root inherited attribute to the representation returned by the encoder for the context information.

**Choosing Productions, Variables & Literals**    We can treat picking production rules as a simple classification problem over all valid production rules, masking out those choices that do not correspond to the currently considered nonterminal. For a nonterminal node $v$ with label $\ell_v$ and inherited attributes $\mathbf{h}_v$, we thus define

$$\mathsf{pickProduction}(\ell_v, \mathbf{h}_v) = \arg\max P(rule \mid \ell_v, \mathbf{h}_v) = \arg\max \left[ e(\mathbf{h}_v) + m_{\ell_v} \right]. \tag{3}$$

Here, $m_{\ell_v}$ is a mask vector whose value is 0 for valid productions $\ell_v \Rightarrow \ldots$ and $-\infty$ for all other productions. In practice, we implement $e$ using a linear layer.

Similarly, we pick variables from the set of variables $\mathcal{V}$ in scope using their representations $\mathbf{h}_{v_{var}}$ (initially the representation obtained from the context, and later the attribute representation of the last node in the graph in which they have been used) by using a pointer network (Vinyals et al., 2015). Concretely, to pick a variable at node $v$, we use learnable linear function $k$ and define

$$\mathsf{pickVariable}(\mathcal{V}, \mathbf{h}_v) = \arg\max_{var \in \mathcal{V}} P(var \mid \mathbf{h}_v) = \arg\max_{var \in \mathcal{V}} k(\mathbf{h}_v, \mathbf{h}_{v_{var}}). \tag{4}$$

Note that since the model always picks a variable from the set of in-scope variables $\mathcal{V}$, this generation model can never predict an unknown or out-of-scope variable.

Finally, to generate literals, we combine a small vocabulary $\mathcal{L}$ of common literals observed in the training data and special UNK tokens for each type of literal with another pointer network that can copy one of the tokens $t_1 \ldots t_T$ from the context. Thus, to pick a literal at node $v$, we define

$$\mathsf{pickLiteral}(\mathcal{V}, \mathbf{h}_v) = \arg\max_{lit \in \mathcal{L} \cup \{t_1 \ldots t_T\}} P(lit \mid \mathbf{h}_v). \tag{5}$$

Note that this is the only operation that may produce an unknown token (*i.e.* an UNK literal). In practice, we implement this by learning two functions $s_{\mathcal{L}}$ and $s_c$, such that $s_{\mathcal{L}}(\mathbf{h}_v)$ produces a score for each token from the vocabulary and $s_c(\mathbf{h}_v, h_{t_i})$ computes a score for copying token $t_i$ from the context. By computing a softmax over all resulting values and normalizing it by summing up entries corresponding to the same constant, we can learn to approximate the desired $P(lit \mid \mathbf{h}_v)$.

**Training & Training Objective** The different shapes and sizes of generated expressions complicate an efficient training regime. However, note that given a ground truth target tree, we can easily augment it with all additional edges according to Alg. 2. Given that full graph, we can compute a propagation schedule (intuitively, a topological ordering of the nodes in the graph, starting in the root node) that allows to repeatedly apply (2) to obtain representations for all nodes in the graph. By representing a batch of graphs as one large (sparse) graph with many disconnected components, similar to Allamanis et al. (2018b), we can train our graph neural network efficiently. We have released the code for this on `https://github.com/Microsoft/graph-based-code-modelling`.

Our training procedure thus combines an encoder (*cf.* Sect. 5), whose output is used to initialize the representation of the root and context variable nodes in our augmented syntax graph, the sequential graph propagation procedure described above, and the decoder choice functions (3) and (4). We train the system end-to-end using a maximum likelihood objective without pre-trained components.

**Additional Improvements** We extend (3) with an attention mechanism (Bahdanau et al., 2014; Luong et al., 2015) that uses the state $\mathbf{h}_v$ of the currently expanded node $v$ as a key and the context token representations $h_{t_1}, \ldots, h_{t_T}$ as memories. Experimentally, we found that extending Eqs. 4, 5 similarly did *not* improve results, probably due to the fact that they already are highly dependent on the context information.

Following Rabinovich et al. (2017), we provide additional information for *Child* edges. To allow this, we change our setup so that some edge types also require an additional label, which is used when computing the messages sent between different nodes in the graph. Concretely, we extend (2) by considering sets of unlabeled edges $\mathcal{E}_v$ and labeled edges $\mathcal{E}_v^\ell$:

$$\mathbf{h}_v = g(\mathsf{emb}(\ell_v), \sum_{(u_i, t_i, v) \in \mathcal{E}_v} f_{t_i}(\mathbf{h}_{u_i}) + \sum_{(u_i, t_i, \ell_i, v) \in \mathcal{E}_v^\ell} f_{t_i}(\mathbf{h}_{u_i}, \mathsf{emb}_e(\ell_i))) \tag{6}$$

Thus for labeled edge types, $f_{t_i}$ takes two inputs and we additionally introduce a learnable embedding for the edge labels. In our experiments, we found it useful to label *Child* with tuples consisting of the chosen production and the index of the child, *i.e.*, in Fig. 2, we would label the edge from 0 to 3 with $(2, 0)$, the edge from 0 to 6 with $(2, 1)$, *etc.*

Furthermore, we have extended pickProduction to also take the information about available variables into account. Intuitively, this is useful in cases of productions such as $\mathtt{Expr} \implies \mathtt{Expr.Length}$, which can only be used in a well-typed derivation if an array-typed variable is available. Thus, we extend $e(\mathbf{h}_v)$ from (3) to additionally take the representation of all variables in scope into account, *i.e.*, $e(\mathbf{h}_v, r(\{\mathbf{h}_{v_{var}} \mid var \in \mathcal{V}\}))$, where we have implemented $r$ as a max pooling operation.

## 4 RELATED WORK

Source code generation has been studied in a wide range of different settings (Allamanis et al., 2018a). We focus on the most closely related works in language modeling here. Early works approach the task by generating code as sequences of tokens (Hindle et al., 2012; Hellendoorn & Devanbu, 2017), whereas newer methods have focused on leveraging the known target grammar and generate code as trees (Maddison & Tarlow, 2014; Bielik et al., 2016; Parisotto et al., 2017; Yin & Neubig, 2017; Rabinovich et al., 2017) (*cf.* Sect. 2 for an overview). While modern models succeed at generating "natural-looking" programs, they often fail to respect simple semantic rules. For example, variables are often used without initialization or written several times without being read inbetween.

Existing tree-based generative models primarily differ in what information they use to decide which expansion rule to use next. Maddison & Tarlow (2014) consider the representation of the immediate parent node, and suggest to consider more information (*e.g.*, nearby tokens). Parisotto et al. (2017) compute a fresh representation of the partial tree at each expansion step using R3NNs (which intuitively perform a leaf-to-root traversal followed by root-to-leaf traversal of the AST). The PHOG model (Bielik et al., 2016) conditions generation steps on the result of learned (decision tree-style) programs, which can do bounded AST traversals to consider nearby tokens and non-terminal nodes. The language also supports a jump to the last node with the same identifier, which can serve as syntactic approximation of data-flow analysis. Rabinovich et al. (2017) only use information about the parent node, but use neural networks specialized to different non-terminals to gain more fine-grained

control about the flow of information to different successor nodes. Finally, Amodio et al. (2017) and Yin & Neubig (2017) follow a left-to-right, depth-first expansion strategy, but thread updates to single state (via a gated recurrent unit) through the overall generation procedure, thus giving the pickProduction procedure access to the full generation history as well as the representation of the parent node. Amodio et al. (2017) also suggest the use of attribute grammars, but use them to define a deterministic procedure that collects information throughout the generation process, which is provided as additional feature.

As far as we are aware, previous work has not considered a task in which a generative model fills a hole in a program with an expression. Lanuage model-like methods take into account only the lexicographically previous context of code. The task of Raychev et al. (2014) is near to our ExprGen, but instead focuses on filling holes in sequences of API calls. There, the core problem is identifying the correct function to call from a potentially large set of functions, given a sequence context. In contrast, ExprGen requires to handle arbitrary code in the context, and then to build possibly complex expressions from a small set of operators. Allamanis et al. (2018b) consider similar context, but are only picking a single variable from a set of candidates, and thus require no generative modeling.

## 5 EVALUATION

**Dataset**  We have collected a dataset for our ExprGen task from 593 highly-starred open-source C$^{\#}$ projects on GitHub, removing any near-duplicate files, following the work of Lopes et al. (2017). We parsed all C$^{\#}$ files and identified all expressions of the fragment that we are considering (*i.e.*, restricted to numeric, Boolean and string types, or arrays of such values; and not using any user-defined functions). We then remove the expression, perform a static analysis to determine the necessary context information and extract a sample. For each sample, we create an abstract syntax tree by coarsening the syntax tree generated by the C$^{\#}$ compiler Roslyn. This resulted in 343 974 samples overall with 4.3 ($\pm 3.8$) tokens per expression to generate, or alternatively 3.7 ($\pm$ 3.1) production steps. We split the data into four separate sets. A "test-only" dataset is made up from $\sim$100k samples generated from 114 projects. The remaining data we split into training-validation-test sets (3 : 1 : 1), keeping all expressions collected from a single source file within a single fold. Samples from our dataset can be found in the supplementary material. Our decoder uses the grammar made up by 222 production rules observed in the ASTs of the training set, which includes rules such as `Expr` $\Longrightarrow$ `Expr + Expr` for binary operations, `Expr` $\Longrightarrow$ `Expr.Equals(Expr)` for built-in methods, *etc.*

**Encoders**  We consider two models to encode context information. $\mathcal{S}eq$ is a two-layer bi-directional recurrent neural network (using a GRU (Cho et al., 2014)) to encode the tokens before and after the "hole" in which we want to generate an expression. Additionally, it computes a representation for each variable $var$ in scope in the context in a similar manner: For each variable $var$ it identifies usages before/after the hole and encodes each of them independently using a second bi-directional two-layer GRU, which processes a window of tokens around each variable usage. It then computes a representation for $var$ by average pooling of the final states of these GRU runs.

The second encoder $\mathcal{G}$ is an implementation of the program graph approach introduced by Allamanis et al. (2018b). We follow the transformation used for the Varmisuse task presented in that paper, *i.e.*, the program is transformed into a graph, and the target expression is replaced by a fresh dummy node. We then run a graph neural network for 8 steps to obtain representations for all nodes in the graph, allowing us to read out a representation for the "hole" (from the introduced dummy node) and for all variables in context. The used context information captured by the GNN is a superset of what existing methods (*e.g.* language models) consider.

**Baseline Decoders**  We compare our model to re-implementations of baselines from the literature. As our ExprGen task is new, re-using existing implementations is hard and problematic in comparison. Most recent baseline methods can be approximated by ablations of our model. We experimented with a simple sequence decoder with attention and copying over the input, but found it to be substantially weaker than other models in all regards. Next, we consider $\mathcal{T}ree$, our model restricted to using only *Child* edges without edge labels. This can be viewed as an evolution of Maddison & Tarlow (2014), with the difference that instead of a log-bilinear network that does not maintain state during the generation, we use a GRU. $\mathcal{ASN}$ is similar to abstract syntax networks (Rabinovich et al., 2017) and arises as an extension of the $\mathcal{T}ree$ model by adding edge labels on *Child* that encode the chosen

Table 1: Evaluation of encoder and decoder combinations on predicting an expression from code context. †: PHOG (Bielik et al., 2016) is only conditioned on the tokens on the left of the expression.

| Model | Test (from seen projects) | | | | Test-only (from unseen projects) | | | |
|---|---|---|---|---|---|---|---|---|
| | Perplexity | Well-Typed | Acc@1 | Acc@5 | Perplexity | Well-Typed | Acc@1 | Acc@5 |
| $PHOG^\dagger$ | – | – | 34.8% | 42.9% | – | – | 28.0% | 37.3% |
| $\mathcal{Seq} \to \mathcal{Seq}$ | 87.48 | 32.4% | 21.8% | 28.1% | 130.46 | 23.4% | 10.8% | 16.8% |
| $\mathcal{Seq} \to \mathcal{NAG}$ | 6.81 | 53.2% | 17.7% | 33.7% | 8.38 | 40.4% | 8.4% | 15.8% |
| $\mathcal{G} \to \mathcal{Seq}$ | 93.31 | 40.9% | 27.1% | 34.8% | 28.48 | 36.3% | 17.2% | 25.6% |
| $\mathcal{G} \to \mathcal{Tree}$ | 4.37 | 49.3% | 26.8% | 48.9% | 5.37 | 41.2% | 19.9% | 36.8% |
| $\mathcal{G} \to \mathcal{ASN}$ | 2.62 | 78.7% | 45.7% | 62.0% | 3.03 | 74.7% | 32.4% | 48.1% |
| $\mathcal{G} \to \mathcal{Syn}$ | 2.71 | 84.9% | 50.5% | 66.8% | 3.48 | 84.5% | 36.0% | 52.7% |
| $\mathcal{G} \to \mathcal{NAG}$ | 2.56 | 86.4% | 52.3% | 69.2% | 3.07 | 84.5% | 38.8% | 57.0% |

production and the index of the child (corresponding to the "field name" Rabinovich et al. (2017)). Finally, $\mathcal{Syn}$ follows the work of Yin & Neubig (2017), but uses a GRU instead of an LSTM. For this, we extend $\mathcal{Tree}$ by a new *NextExp* edge that connects nodes to each other in the expansion sequence of the tree, thus corresponding to the action flow (Yin & Neubig, 2017).

In all cases, our re-implementations improve on prior work in our variable selection mechanism, which ensures that generated programs only use variables that are defined and in scope. Both Rabinovich et al. (2017) and Yin & Neubig (2017) instead use a copying mechanism from the context. On the other hand, they use RNN modules to generate function names and choose arguments from the context (Yin & Neubig, 2017) and to generate string literals (Rabinovich et al., 2017). Our ExprGen task limits the set of allowed functions and string literals substantially and thus no RNN decoder generating such things is required in our experiments.

The authors of the PHOG (Bielik et al., 2016) language model kindly ran experiments on our data for the ExprGen task, to provide baseline results of a non-neural language model. Note, however, that PHOG does not consider the code context to the right of the expression to generate, and does no additional analyses to determine which variable choices are valid. Extending the model to take more context into account and do some analyses to restrict choices would certainly improve its results.

### 5.1 QUANTITATIVE EVALUATION

**Metrics** We are interested in the ability of a model to generate valid expressions based on the current code context. To evaluate this, we consider four metrics. As our ExprGen task requires a conditional language model of code, we first consider the per-token perplexity of the model; the lower the perplexity, the better the model fits the real data distribution. We then evaluate how often the generated expression is well-typed (*i.e.*, can be typed in the original code context). We report these metrics for the most likely expression returned by beam search decoding with beam width 5. Finally, we compute how often the ground truth expression was generated (reported for the most likely expression, as well as for the top five expressions). This measure is stricter than semantic equivalence, as an expression `j > i` will not match the equivalent `i < j`.

**Results** We show the results of our evaluation in Tab. 1. Overall, the graph encoder architecture seems to be best-suited for this task. All models learn to generate syntactically valid code (which is relatively simple in our domain). However, the different encoder models perform very differently on semantic measures such as well-typedness and the retrieval of the ground truth expression. Most of the type errors are due to usage of an "UNK" literal (for example, the $\mathcal{G} \to \mathcal{NAG}$ model only has 4% type error when filtering out such unknown literals). The results show a clear trend that correlates better semantic results with the amount of information about the partially generated programs employed by the generative models. Transferring a trained model to unseen projects with a new project-specific vocabulary substantially worsens results, as expected. Overall, our $\mathcal{NAG}$ model, combining and adding additional signal sources, seems to perform best on most measures, and seems to be least-impacted by the transfer.

```
int methParamCount = 0;
if (paramCount > 0) {
  IParameterTypeInformation[] moduleParamArr =
    GetParamTypeInformations(Dummy.Signature, paramCount);
  methParamCount = moduleParamArr.Length;
}

if ( paramCount > methParamCount ) {
  IParameterTypeInformation[] moduleParamArr =
    GetParamTypeInformations(Dummy.Signature,
                       paramCount - methParamCount);
}
```

$\mathcal{G} \to \mathcal{NAG}$:
paramCount > methParamCount (34.4%)
paramCount == methParamCount (11.4%)
paramCount < methParamCount (10.0%)

$\mathcal{G} \to \mathcal{ASN}$:
paramCount == 0 (12.7%)
paramCount < 0 (11.5%)
paramCount > 0 (8.0%)

```
public static String URItoPath(String uri) {
    if (System.Text.RegularExpressions
          .Regex.IsMatch(uri, "^file:\\\\[a-z,A-Z]:")) {
        return uri.Substring(6);
    }
    if ( uri.StartsWith(@"file:") ) {
        return uri.Substring(5);
    }
    return uri;
}
```

$\mathcal{G} \to \mathcal{NAG}$:
uri.Contains(UNK_STRING_LITERAL) (32.4%)
uri.StartsWith(UNK_STRING_LITERAL) (29.2%)
uri.HasValue() (7.7%)

$\mathcal{G} \to \mathcal{S}yn$:
uri == UNK_STRING_LITERAL (26.4%)
uri == "" (8.5%)
uri.StartsWith(UNK_STRING_LITERAL) (6.7%)

Figure 3: Two lightly edited examples from our test set and expressions predicted by different models. More examples can be found in the supplementary material.

## 5.2 QUALITATIVE EVALUATION

As the results in the previous section suggest, the proposed ExprGen task is hard even for the strongest models we evaluated, achieving no more than 50% accuracy on the top prediction. It is also unsolvable for classical logico-deductive program synthesis systems, as the provided code context does not form a precise specification. However, we do know that most instances of the task are (easily) solvable for professional software developers, and thus believe that machine learning systems can have considerable success on the task.

Fig. 3 shows two (abbreviated) samples from our test set, together with the predictions made by the two strongest models we evaluated. In the first example, we can see that the $\mathcal{G} \to \mathcal{NAG}$ model correctly identifies that the relationship between paramCount and methParamCount is important (as they appear together in the blocked guarded by the expression to generate), and thus generates comparison expressions between the two variables. The $\mathcal{G} \to \mathcal{ASN}$ model lacks the ability to recognize that paramCount (or any variable) was already used and thus fails to insert both relevant variables. We found this to be a common failure, often leading to suggestions using only one variable (possibly repeatedly). In the second example, both $\mathcal{G} \to \mathcal{NAG}$ and $\mathcal{G} \to \mathcal{S}yn$ have learned the common if (var.StartsWith(...)) { ... var.Substring(num) ... } pattern, but of course fail to produce the correct string literal in the condition. We show results for all of our models for these examples, as well as for as additional examples, in the supplementary material B.

## 6 DISCUSSION & CONCLUSIONS

We presented a generative code model that leverages known semantics of partially generated programs to direct the generative procedure. The key idea is to augment partial programs to obtain a graph, and then use graph neural networks to compute a precise representation for the partial program. This representation then helps to better guide the remainder of the generative procedure. We have shown that this approach can be used to generate small but semantically interesting expressions from very imprecise context information. The presented model could be useful in program repair scenarios (where repair proposals need to be scored, based on their context) or in the code review setting (where it could highlight very unlikely expressions). We also believe that similar models could have applications in related domains, such as semantic parsing, neural program synthesis and text generation.

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

## A   DATASET SAMPLES

Below we list some sample snippets from the training set for our ExprGen task. The highlighted expressions are to be generated.

```
for (int i=0; i < 3*timeSpanUnits + 1 ; ++i) {
   consolidator.Update(new TradeBar { Time = refDateTime });

   if (i < timeSpanUnits) { // before initial consolidation happens
      Assert.IsNull(consolidated);
   }
   else {
      Assert.IsNotNull(consolidated);
      if ( i % timeSpanUnits == 0 ) { // i = 3, 6, 9
         Assert.AreEqual(refDateTime.AddMinutes(-timeSpanUnits), consolidated.Time);
      }
   }

   refDateTime = refDateTime.AddMinutes(1);
}
```

Figure 4: Sample snippet from the Lean project. Formatting has been modified.

```
var words = (from word in  phrase.Split(' ')
             where word.Length > 0 select word.ToLower()).ToArray();
```

Figure 5: Sample snippet from the BotBuilder project. Formatting has been modified.

```
_hasHandle = _mutex.WaitOne( timeOut < 0  ? Timeout.Infinite
                                   : timeOut,
                      exitContext: false);
```

Figure 6: Sample snippet from the Chocolatey project. Formatting has been modified.

```
public static T retry<T>(int numberOfTries, Func<T> function,
                    int waitDurationMilliseconds = 100,
                    int increaseRetryByMilliseconds = 0) {
   if (function == null) return default(T);
   if (numberOfTries == 0)
      throw new ApplicationException("You must specify a number"
                               + " of retries greater than zero.");
   var returnValue = default(T);

   var debugging = log_is_in_debug_mode();
   var logLocation = ChocolateyLoggers.Normal;

   for (int i = 1;  i <= numberOfTries ; i++)
   {
```

Figure 7: Sample snippet from the Chocolatey project. Formatting has been modified and the snippet has been abbreviated.

```
while ( count >= startIndex )
{
    c = s[count];
    if ( c != ' ' && c != 'n' ) break;
    count--;
}
```

Figure 8: Samples snippet in the CommonMark.NET project. Formatting has been modified.

```
private string GetResourceForTimeSpan(TimeUnit unit, int count)
{
    var resourceKey = ResourceKeys.TimeSpanHumanize.GetResourceKey(unit, count);
    return  count == 1  ? Format(resourceKey) : Format(resourceKey, count);
}
```

Figure 9: Sample snippet from the Humanizer project. Formatting has been modified.

```
var indexOfEquals =  segment.IndexOf('=') ;
if ( indexOfEquals == -1 ) {
    var decoded = UrlDecode(segment, encoding);
    return new KeyValuePair<string, string>(decoded, decoded);
}
```

Figure 10: Samples snippet from the Nancy project. Formatting has been modified.

```
private bool ResolveWritableOverride(bool writable)
{
    if (!Writable && writable)
        throw new StorageInvalidOperationException("Cannot open writable storage"
                                            + " in readonly storage.");

    bool openWritable = Writable;
    if ( openWritable && !writable )
        openWritable = writable;
    return openWritable;
}
```

Figure 11: Sample snippet from the OpenLiveWriter project. Formatting has been modified.

```
char c = html[j];
if ( c == ';' || (!(c >= 'a' && c <= 'z') && !(c >= 'A' && c <= 'Z') && !(c >= '0' && c <= '9')) )
{
    break;
}
```

Figure 12: Sample snippet from the OpenLiveWriter project. Formatting has been modified.

```
string entityRef = html.Substring(i + 1, j - (i + 1));
```

Figure 13: Sample snippet from the OpenLiveWriter project. Formatting has been modified.

## B  SAMPLE GENERATIONS

On the following pages, we list some sample snippets from the test set for our ExprGen task, together with suggestions produced by different models. The highlighted expressions are the ground truth expression that should be generated.

**Sample 1**

```
if (context.Context == _MARKUP_CONTEXT_TYPE.CONTEXT_TYPE_Text &&
        !String.IsNullOrEmpty(text)) {
    idx = originalText.IndexOf(text) ;
    if (idx == 0) {
        // Drop this portion from the expected string
        originalText = originalText.Substring(text.Length);

        // Update the current pointer
        beginDamagePointer.MoveToPointer(currentRange.End);
    }
    else if (idx > 0 &&
        originalText.Substring(0, idx)
            .Replace("\r\n", string.Empty).Length == 0)
    {
        // Drop this portion from the expected string
        originalText = originalText.Substring(text.Length + idx);
        // Update the current pointer
        beginDamagePointer.MoveToPointer(currentRange.End);
    }
    else
    {
        return false;
    }
}
```

Sample snippet from OpenLiveWriter. The following suggestions were made:

$\mathcal{Seq} \rightarrow \mathcal{Seq}$:

```
UNK_TOKEN[i] (0.6%)
input[inputOffset + 1] (0.3%)
UNK_TOKEN & UNK_NUM_LITERAL (0.3%)
```

$\mathcal{Seq} \rightarrow \mathcal{NAG}$:

```
MarshalUrlSupported.IndexOf(UNK_CHAR_LITERAL) (0.9%)
IsEditFieldSelected.IndexOf(UNK_CHAR_LITERAL) (0.8%)
marshalUrlSupported.IndexOf(UNK_CHAR_LITERAL) (0.7%)
```

$\mathcal{G} \rightarrow \mathcal{Seq}$:

```
UNK_TOKEN.IndexOf(UNK_CHAR_LITERAL) (21.6%)
UNK_TOKEN.LastIndexOf(UNK_CHAR_LITERAL) (14.9%)
UNK_TOKEN.GetHashCode() (8.1%)
```

$\mathcal{G} \rightarrow \mathcal{Tree}$:

```
UNK_CHAR_LITERAL.IndexOf(UNK_CHAR_LITERAL) (8.1%)
UNK_CHAR_LITERAL.IndexOf(originalText) (8.1%)
originalText.IndexOf(UNK_CHAR_LITERAL) (8.1%)
```

$\mathcal{G} \rightarrow \mathcal{ASN}$:

```
originalText.GetHashCode() (37.8%)
originalText.IndexOf(UNK_CHAR_LITERAL) (14.8%)
originalText.LastIndexOf(UNK_CHAR_LITERAL) (6.2%)
```

$\mathcal{G} \rightarrow \mathcal{Syn}$:

```
text.IndexOf(UNK_CHAR_LITERAL) (20.9%)
text.LastIndexOf(UNK_CHAR_LITERAL) (12.4%)
originalText.IndexOf(UNK_CHAR_LITERAL) (11.6%)
```

$\mathcal{G} \rightarrow \mathcal{NAG}$:

```
originalText.IndexOf(UNK_CHAR_LITERAL) (32.8%)
originalText.LastIndexOf(UNK_CHAR_LITERAL) (12.4%)
originalText.IndexOf(text) (8.7%)
```

**Sample 2**

```
caretPos--;
if (caretPos < 0) {
    caretPos = 0;
}

int len = inputString.Length;
if (caretPos >= len) {
    caretPos = len - 1 ;
}
```

Sample snippet from acat. The following suggestions were made:

$\mathcal{Seq} \to \mathcal{Seq}$:

UNK_TOKEN+1 (2.1%)

UNK_TOKEN+UNK_TOKEN] (1.8%)

UNK_TOKEN.IndexOf(UNK_CHAR_LITERAL) (1.3%)

$\mathcal{Seq} \to \mathcal{NAG}$:

wordToReplace - 1 (3.2%)

insertOrReplaceOffset - 1 (2.9%)

inputString - 1 (1.9%)

$\mathcal{G} \to \mathcal{Seq}$:

len + 1 (35.6%)

len - 1 (11.3%)

len >> UNK_NUM_LITERAL (3.5%)

$\mathcal{G} \to \mathcal{Tree}$:

len + len (24.9%)

len - len (10.7%)

1 + len (3.7%)

$\mathcal{G} \to \mathcal{ASN}$:

len + 1 (22.8%)

len - 1 (10.8%)

len + len (10.3%)

$\mathcal{G} \to \mathcal{Syn}$:

len + 1 (13.7%)

len - 1 (11.5%)

len - len (11.0%)

$\mathcal{G} \to \mathcal{NAG}$:

len++ (33.6%)

len-1 (21.9%)

len+1 (14.6%)

**Sample 3**

```
public static String URItoPath(String uri) {
    if (System.Text.RegularExpressions
            .Regex.IsMatch(uri, "^file:\\\\[a-z,A-Z]:")) {
        return uri.Substring(6);
    }
    if ( uri.StartsWith(@"file:") ) {
        return uri.Substring(5);
    }
    return uri;
}
```

Sample snippet from acat. The following suggestions were made:

$\mathcal{Seq} \to \mathcal{Seq}$:
!UNK_TOKEN (11.1%)
UNK_TOKEN == 0 (3.6%)
UNK_TOKEN != 0 (3.4%)

$\mathcal{Seq} \to \mathcal{NAG}$:
!uri (7.6%)
!MyVideos (4.7%)
!MyDocuments (4.7%)

$\mathcal{G} \to \mathcal{Seq}$:
action == UNK_STRING_LITERAL (22.6%)
label == UNK_STRING_LITERAL (14.8%)
file.Contains(UNK_STRING_LITERAL) (4.6%)

$\mathcal{G} \to \mathcal{Tree}$:
uri == uri (7.4%)
uri.StartsWith(uri) (5.5%)
uri.Contains(uri) (4.3%)

$\mathcal{G} \to \mathcal{ASN}$:
uri == UNK_STRING_LITERAL (11.7%)
uri.Contains(UNK_STRING_LITERAL) (11.7%)
uri.StartsWith(UNK_STRING_LITERAL) (8.3%)

$\mathcal{G} \to \mathcal{Syn}$:
uri == UNK_STRING_LITERAL (26.4%)
uri == "" (8.5%)
uri.StartsWith(UNK_STRING_LITERAL) (6.7%)

$\mathcal{G} \to \mathcal{NAG}$:
uri.Contains(UNK_STRING_LITERAL) (32.4%)
uri.StartsWith(UNK_STRING_LITERAL) (29.2%)
uri.HasValue() (7.7%)

**Sample 4**

```
startPos = index + 1;
int count = endPos - startPos + 1;

word = (count > 0) ? input.Substring(startPos, count) : String.Empty;
```

Sample snippet from acat. The following suggestions were made:

$\mathcal{S}eq \rightarrow \mathcal{S}eq$:

UNK_TOKEN.Trim() (3.4%)

UNK_TOKEN.Replace(UNK_STRING_LITERAL, UNK_STRING_LITERAL) (2.1%)

UNK_TOKEN.Replace('UNK_CHAR', 'UNK_CHAR') (3.4%)

$\mathcal{S}eq \rightarrow \mathcal{NAG}$:

input[index] (1.4%)

startPos[input] (0.9%)

input[count] (0.8%)

$\mathcal{G} \rightarrow \mathcal{S}eq$:

val.Trim() (6.6%)

input.Trim() (6.5%)

input.Substring(UNK_NUM_LITERAL) (4.0%)

$\mathcal{G} \rightarrow \mathcal{T}ree$:

UNK_STRING_LITERAL + UNK_STRING_LITERAL (8.4%)

UNK_STRING_LITERAL + startPos (7.8%)

startPos + UNK_STRING_LITERAL (7.8%)

$\mathcal{G} \rightarrow \mathcal{ASN}$:

input.Trim() (15.6%)

input.Substring(0) (6.4%)

input.Replace(UNK_STRING_LITERAL, UNK_STRING_LITERAL) (2.8%)

$\mathcal{G} \rightarrow \mathcal{S}yn$:

input.Trim() (7.8%)

input.ToLower() (6.4%)

input + UNK_STRING_LITERAL (5.6%)

$\mathcal{G} \rightarrow \mathcal{NAG}$:

input+StartPos (11.8%)

input+count (9.5%)

input.Substring(startPos, endPos - count) (6.3%)

**Sample 5**

```
protected virtual void CrawlSite() {
    while ( !_crawlComplete )
    {
        RunPreWorkChecks();

        if (_scheduler.Count > 0) {
            _threadManager.DoWork(
                () => ProcessPage(_scheduler.GetNext()));
        }
        else if (!_threadManager.HasRunningThreads()) {
            _crawlComplete = true;
        } else {
            _logger.DebugFormat("Waiting for links to be scheduled...");
            Thread.Sleep(2500);
        }
    }
}
```

Sample snippet from Abot. The following suggestions were made:

$\mathcal{S}eq \rightarrow \mathcal{S}eq$:

`!UNK_TOKEN` (9.4%)
`UNK_TOKEN > 0` (2.6%)
`UNK_TOKEN != value` (1.3%)

$\mathcal{S}eq \rightarrow \mathcal{NAG}$:

`!_maxPagesToCrawlLimitReachedOrScheduled` (26.2%)
`!_crawlCancellationReported` (26.0%)
`!_crawlStopReported` (21.8%)

$\mathcal{G} \rightarrow \mathcal{S}eq$:

`!UNK_TOKEN` (54.9%)
`!done` (18.8%)
`!throwOnError` (3.3%)

$\mathcal{G} \rightarrow \mathcal{T}ree$:

`!_crawlCancellationReported` (23.6%)
`!_crawlStopReported` (23.3%)
`!_maxPagesToCrawlLimitReachedOrScheduled` (18.9%)

$\mathcal{G} \rightarrow \mathcal{ASN}$:

`!_crawlStopReported` (26.6%)
`!_crawlCancellationReported` (26.5%)
`!_maxPagesToCrawlLimitReachedOrScheduled` (25.8%)

$\mathcal{G} \rightarrow \mathcal{S}yn$:

`!_crawlStopReported` (19.6%)
`!_maxPagesToCrawlLimitReachedOrScheduled` (19.0%)
`!_crawlCancellationReported` (15.7%)

$\mathcal{G} \rightarrow \mathcal{NAG}$:

`!_crawlStopReported` (38.4%)
`!_crawlCancellationReported` (31.8%)
`!_maxPagesToCrawlLimitReachedOrScheduled` (27.0%)

**Sample 6**

```
char character = originalName[i];
if ( character == '<' ) {
    ++startTagCount;
    builder.Append(' ');
} else if (startTagCount > 0) {
    if (character == '>') {
        --startTagCount;
    }
```

Sample snippet from StyleCop. The following suggestions were made:

$\mathcal{S}eq \to \mathcal{S}eq$:

```
x == UNK_CHAR_LITERAL (5.9%)
UNK_TOKEN == 0 (3.3%)
UNK_TOKEN > 0 (2.7%)
```

$\mathcal{S}eq \to \mathcal{NAG}$:

```
!i == 0 (5.1%)
character < 0 (2.7%)
character (2.2%)
```

$\mathcal{G} \to \mathcal{S}eq$:

```
character == UNK_CHAR_LITERAL (70.8%)
character == UNK_CHAR_LITERAL || character == UNK_CHAR_LITERAL (5.8%)
character != UNK_CHAR_LITERAL (3.1%)
```

$\mathcal{G} \to \mathcal{T}ree$:

```
character == character (9.9%)
UNK_CHAR_LITERAL == character (8.2%)
character == UNK_CHAR_LITERAL (8.2%)
```

$\mathcal{G} \to \mathcal{ASN}$:

```
character == UNK_CHAR_LITERAL (43.4%)
character || character (3.3%)
character == UNK_CHAR_LITERAL == UNK_CHAR_LITERAL (3.0%)
```

$\mathcal{G} \to \mathcal{S}yn$:

```
character == UNK_CHAR_LITERAL (39.6%)
character || character == UNK_STRING_LITERAL (5.2%)
character == UNK_STRING_LITERAL (2.8%)
```

$\mathcal{G} \to \mathcal{NAG}$:

```
character == UNK_CHAR_LITERAL (75.5%)
character == '' (2.6%)
character != 'UNK_CHAR (2.5%)
```

**Sample 7**

```
public void AllowAccess(string path)
{
    if (path == null) throw new ArgumentNullException("path");
    if ( !path.StartsWith(" /") )
        throw new ArgumentException(
            string.Format(
                "The path \"{0}\" is not application relative."
                + " It must start with \"~/\".",
                path),
            "path");

    paths.Add(path);
}
```

Sample snippet from cassette. The following suggestions were made:

$\mathcal{Seq} \rightarrow \mathcal{Seq}$:

`UNK_TOKEN < 0` (14.6%)

`!UNK_TOKEN` (7.5%)

`UNK_TOKEN == 0` (3.3%)

$\mathcal{Seq} \rightarrow \mathcal{NAG}$:

`path == UNK_STRING_LITERAL` (18.1%)

`path <= 0` (5.6%)

`path == ""` (4.8%)

$\mathcal{G} \rightarrow \mathcal{Seq}$:

`!UNK_TOKEN` (48.0%)

`!discardNulls` (6.3%)

`!first` (2.7%)

$\mathcal{G} \rightarrow \mathcal{Tree}$:

`!path` (67.4%)

`path && path` (8.4%)

`!!path` (5.5%)

$\mathcal{G} \rightarrow \mathcal{ASN}$:

`!path` (91.5%)

`!path && !path` (0.9%)

`!path.Contains(UNK_STRING_LITERAL)` (0.7%)

$\mathcal{G} \rightarrow \mathcal{Syn}$:

`!path` (89.6%)

`!path && !path` (1.5%)

`!path.Contains(UNK_STRING_LITERAL)` (0.5%)

$\mathcal{G} \rightarrow \mathcal{NAG}$:

`!path` (42.9%)

`!path.StartsWith(UNK_STRING_LITERAL)` (23.8%)

`!path.Contains(UNK_STRING_LITERAL)` (5.9%)

**Sample 8**

```
int methodParamCount = 0;
IEnumerable<IParameterTypeInformation> moduleParameters =
    Enumerable<IParameterTypeInformation>.Empty;
if (paramCount > 0) {
    IParameterTypeInformation[] moduleParameterArr =
        this.GetModuleParameterTypeInformations(Dummy.Signature, paramCount);
    methodParamCount = moduleParameterArr.Length;
    if (methodParamCount > 0)
        moduleParameters = IteratorHelper.GetReadonly(moduleParameterArr);
}
IEnumerable<IParameterTypeInformation> moduleVarargsParameters =
                    Enumerable<IParameterTypeInformation>.Empty;
if ( paramCount > methodParamCount ) {
    IParameterTypeInformation[] moduleParameterArr =
        this.GetModuleParameterTypeInformations(
            Dummy.Signature, paramCount - methodParamCount);
    if (moduleParameterArr.Length > 0)
        moduleVarargsParameters = IteratorHelper.GetReadonly(moduleParameterArr);
}
```

Sample snippet from Afterthought. The following suggestions were made:

$\mathcal{S}eq \rightarrow \mathcal{S}eq$:

`!UNK_TOKEN` (10.9%)

`UNK_TOKEN == UNK_TOKEN` (4.6%)

`UNK_TOKEN == UNK_STRING_LITERAL` (3.3%)

$\mathcal{S}eq \rightarrow \mathcal{N}\mathcal{A}\mathcal{G}$:

`dummyPinned != 0` (2.2%)

`paramCount != 0` (2.1%)

`dummyPinned == 0` (1.5%)

$\mathcal{G} \rightarrow \mathcal{S}eq$:

`newValue > 0` (9.7%)

`zeroes > 0` (9.0%)

`paramCount > 0` (6.0%)

$\mathcal{G} \rightarrow \mathcal{T}ree$:

`methodParamCount == methodParamCount` (3.4%)

`0 == methodParamCount` (2.8%)

`methodParamCount == paramCount` (2.8%)

$\mathcal{G} \rightarrow \mathcal{A}\mathcal{S}\mathcal{N}$:

`paramCount == 0` (12.7%)

`paramCount < 0` (11.5%)

`paramCount > 0` (8.0%)

$\mathcal{G} \rightarrow \mathcal{S}yn$:

`methodParamCount > 0` (10.9%)

`paramCount > 0` (7.9%)

`methodParamCount != 0` (5.6%)

$\mathcal{G} \rightarrow \mathcal{N}\mathcal{A}\mathcal{G}$:

`paramCount > methodParamCount` (34.4%)

`paramCount == methodParamCount` (11.4%)

`paramCount < methodParamCount` (10.0%)

**Sample 9**

```
public CodeLocation(int index, int endIndex, int indexOnLine,
               int endIndexOnLine, int lineNumber, int endLineNumber)
{
    Param.RequireGreaterThanOrEqualToZero(index, "index");
    Param.RequireGreaterThanOrEqualTo(endIndex, index, "endIndex");
    Param.RequireGreaterThanOrEqualToZero(indexOnLine, "indexOnLine");
    Param.RequireGreaterThanOrEqualToZero(endIndexOnLine, "endIndexOnLine");
    Param.RequireGreaterThanZero(lineNumber, "lineNumber");
    Param.RequireGreaterThanOrEqualTo(endLineNumber, lineNumber, "endLineNumber");

    // If the entire segment is on the same line,
    // make sure the end index is greater or equal to the start index.
    if ( lineNumber == endLineNumber ) {
        Debug.Assert(endIndexOnLine >= indexOnLine,
            "The end index must be greater than the start index,"
            + " since they are both on the same line.");
    }

    this.startPoint = new CodePoint(index, indexOnLine, lineNumber);
    this.endPoint = new CodePoint(endIndex, endIndexOnLine, endLineNumber);
}
```

Sample snippet from StyleCop. The following suggestions were made:

$\mathcal{Seq} \rightarrow \mathcal{Seq}$:

`!UNK_TOKEN` (14.0%)

`UNK_TOKEN == 0` (4.4%)

`UNK_TOKEN > 0` (3.5%)

$\mathcal{Seq} \rightarrow \mathcal{NAG}$:

`endIndex < 0` (3.8%)

`endIndex > 0` (3.4%)

`endIndex == 0` (2.2%)

$\mathcal{G} \rightarrow \mathcal{Seq}$:

`lineNumber < 0` (9.4%)

`lineNumber == 0` (7.4%)

`lineNumber <= 0` (5.1%)

$\mathcal{G} \rightarrow \mathcal{Tree}$:

`lineNumber == lineNumber` (3.4%)

`0 == lineNumber` (2.5%)

`lineNumber > lineNumber` (2.5%)

$\mathcal{G} \rightarrow \mathcal{ASN}$:

`endLineNumber == 0` (9.6%)

`endLineNumber < 0` (7.9%)

`endLineNumber > 0` (6.1%)

$\mathcal{G} \rightarrow \mathcal{Syn}$:

`lineNumber > 0` (11.3%)

`lineNumber == 0` (7.3%)

`lineNumber != 0` (6.7%)

$\mathcal{G} \rightarrow \mathcal{NAG}$:

`lineNumber > endLineNumber` (20.7%)

`lineNumber < endLineNumber` (16.5%)

`lineNumber == endLineNumber` (16.2%)

**Sample 10**

```
public static Bitmap RotateImage(Image img, float angleDegrees,
                         bool upsize, bool clip) {
    // Test for zero rotation and return a clone of the input image
    if (angleDegrees == 0f) return (Bitmap)img.Clone();

    // Set up old and new image dimensions, assuming upsizing not wanted
    // and clipping OK
    int oldWidth = img.Width; int oldHeight = img.Height;
    int newWidth = oldWidth; int newHeight = oldHeight;
    float scaleFactor = 1f;

    // If upsizing wanted or clipping not OK calculate the size of the
    // resulting bitmap
    if ( upsize || !clip ) {
        double angleRadians = angleDegrees * Math.PI / 180d;
        double cos = Math.Abs(Math.Cos(angleRadians));
        double sin = Math.Abs(Math.Sin(angleRadians));
        newWidth = (int)Math.Round((oldWidth * cos) + (oldHeight * sin));
        newHeight = (int)Math.Round((oldWidth * sin) + (oldHeight * cos));
    }
    // If upsizing not wanted and clipping not OK need a scaling factor
    if (!upsize && !clip) {
        scaleFactor = Math.Min((float)oldWidth / newWidth,
                         (float)oldHeight / newHeight);
        newWidth = oldWidth; newHeight = oldHeight;
    }
```

Sample snippet from ShareX. The following suggestions were made:

$\mathcal{Seq} \rightarrow \mathcal{Seq}$:

```
UNK_TOKEN > 0 (8.3%)
!UNK_TOKEN (4.4%)
UNK_TOKEN == 0 (2.6%)
```

$\mathcal{Seq} \rightarrow \mathcal{NAG}$:

```
newHeight > 0 (5.1%)
clip > 0 (3.2%)
oldWidth > 0 (2.9%)
```

$\mathcal{G} \rightarrow \mathcal{Seq}$:

```
UNK_TOKEN && UNK_TOKEN (15.0%)
UNK_TOKEN || UNK_TOKEN (13.6%)
trustedForDelegation && !appOnly (12.1%)
```

$\mathcal{G} \rightarrow \mathcal{Tree}$:

```
upsize && upsize (21.5%)
upsize && clip (10.9%)
clip && upsize (10.9%)
```

$\mathcal{G} \rightarrow \mathcal{ASN}$:

```
upsize && clip (13.9%)
upsize && !clip (9.8%)
clip && clip (9.3%)
```

$\mathcal{G} \rightarrow \mathcal{Syn}$:

```
upsize && !upsize (6.9%)
clip && !upsize (6.3%)
upsize || upsize (5.7%)
```

$\mathcal{G} \rightarrow \mathcal{NAG}$:

```
upsize || clip (19.1%)
upsize && clip (18.8%)
upsize && ! clip (12.2%)
```

