# OpenReview forum: "Generative Code Modeling with Graphs"
_ICLR.cc/2019/Conference_

### Official Review · AnonReviewer3 · 2018-11-01
**Overall good ideas, but not quite ready**

**Rating:** 7
**Confidence:** 5

**Review:**

The paper proposes a code completion task that given the rest of a program, predicts the content of an expression. This task has similarity to code completion tasks in the code editor of an IDE. The paper proposes an interesting problem, but the paper would benefit if writing and evaluation are significantly improved.

The work builds on prior research by Allamanis et al. 2018b that performs such completions of single variables by picking from the variables in the scopes. The difference here is that portions of parse trees are predicted as opposed to a single variables, where the algorithm from the prior research is used to predict single variables.

Writing-wise the paper is hard to read on the technical part with many unclear details and this portion needs a good amount of extra explanations. The Epsilon set includes triples which are not described and need understanding equation (2). The first element of this triple is an edge label <edge>($a$, $v$) where $a$ is an AST and $v$ is a node. Thus, edges of the graph end up between entire ASTs and nodes? While I can see how could this make sense, there is certainly lack of explanation going on here. Overall, this part is hard to parse and time-consuming to understand except at high level. Furthermore, the text has many functions without signatures and they seem to be used before they are defined (e.g. getRepresentation).

Technically, the approach also seems very similar to N3NN by Parisotto et al, ICLR 2017. There should be more elaboration on what is new here. Otherwise, the novelty of the paper really is just combining this work with Allamanis et al. 2018b.

In terms of evaluation, the task seems to be on a different set of expressions than the one explained in the exposition. How many expressions where there in the evaluation programs and how many were chosen to evaluate on and based on what criteria. It seems from the exposition that expressions with field accessed and function calls are not possible to be generated, but then some completions show method calls. How much of the full task is actually solved? In particular, several of the cited prior works solve specific problems like constants that are ignored here.

The evaluation is mostly an ablation studies of the proposed approach by removing edges from the final idea.
Besides this, the paper also introduces a new dataset for showcasing the technique and does not report sizes and running times, essentially not answering basic questions like what is the trade-off between the different techniques. Comparison to actual prior works on similar tasks is also lacking (some TODO is left in the paper), but there is the claim that existing neural techniques such as seq2seq perform "substantially worse". I guess the authors have extra experiments not included for lack of space or that the evaluation was not ready at submission time.

---

> ### Author Response · Authors · 2018-11-15
> **Clarification and answers to review questions**
>
> Thanks for the thorough review. We will try to improve the writing to make the parts of the paper you found hard to follow easier to read.
>
> > The work builds on prior research by Allamanis et al. 2018b that
> > performs such completions of single variables by picking from the
> > variables in the scopes. The difference here is that portions of parse
> > trees are predicted as opposed to a single variables, where the
> > algorithm from the prior research is used to predict single variables.
>
> We want to point out that this is not quite precise -- the model from Allamanis et al. 2018b applies a much more complex analysis to identify the correct variable, introducing speculative data flow edges (i.e., "how would the graph look like if a certain variable were used in this location"). Our method is much more simple, and is more akin to using a pointer network to select a variable available in scope.
>
> > Writing-wise the paper is hard to read on the technical part with many
> > unclear details and this portion needs a good amount of extra
> > explanations. The Epsilon set includes triples which are not described
> > and need understanding equation (2). The first element of this triple
> > is an edge label <edge>($a$, $v$) where $a$ is an AST and $v$ is a
> > node.
>
> You seem to be confusing functions $F(a, v)$ and edges $(u, label, v)$, which are not the same thing and should not be used interchangeably. A function like “parent(a, v)” returns a node from the partial AST $a$ (in this case, the parent node of $v$). An edge is a triple of “(node, label, node)”, explicitly defined in the paragraph right above Eq. (2). The edge labels and function names are (generally) not shared.
>
> We apologize for this confusion. We have updated the paper with a new Notation paragraph that formally defines the constituents of edges triples and functions.
>
> > Thus, edges of the graph end up between entire ASTs and nodes?
>
> No, edges are always between nodes. The functions like parent(a, v) return a /node/ from the partial AST $a$ (e.g. in this case, the parent node of $v$), and should not be confused with edge types.
>
> > Furthermore, the text has many functions without signatures and they
> > seem to be used before they are defined (e.g. getRepresentation).
>
> We indeed stripped the text of explicit signatures for space reasons (as we felt they were implicitly defined anyway), but we remedied this somewhat in the new revision. We have added a Notation paragraph to explain the used functions, which we hope is sufficient. If you feel more context is needed, we can also include explicit signatures everywhere.
>
> > Technically, the approach also seems very similar to N3NN by Parisotto
> > et al, ICLR 2017. There should be more elaboration on what is new
> > here.
>
> [We assume this was a typo, and you refer to R3NNs] The core difference is that R3NNs use only the tree structure. While their up-then-down recursion scheme allows information sharing between different sibling subtrees in principle, no explicit domain knowledge is integrated to directly connect relevant parts of the tree. Our core contribution is to show how to integrate richer domain knowledge directly into the model.
>
> Using a R3NNs in a generative procedure also implies a quadratic computational cost, as each partial tree is traversed twice at each expansion step (summing up to roughly \sum_{1 < i < V} 2i = V^2 + V), whereas our sequential graph propagation requires only a linear pass over all nodes in our graph, where each node in the expansion tree is visited at most twice (once for inherited and once for synthesized attributes).
>
> [On a side note, the authors of the R3NN paper have communicated to us privately that training the model was extremely hard, requiring careful tuning of hyperparameters to ensure convergence to a reasonable state.
> In contrast, our model required almost no hyperparameter tuning to get good results, and we only did a cursory exploration to create the experiments in the paper]

---

> > ### Comment · AnonReviewer3 · 2018-11-16
> > **Clarifications**
> >
> > - confusing functions $F(a, v)$ and edges $(u, label, v)$, which are not the same thing
> >
> > Thank you for the clarification. The problem with $F(a, v)$ being a function is that it is not clear if these functions return a single node (as not all edges have out-degree 1). For example $lastUse(a, v)$ should probably return a set of nodes, because of loops and ifs. However, the new exposition is much better for this.
> >
> > What does inheritedAttr return?
> >
> > - R3NNs
> >
> > Thank you for the clarification. Indeed the proposed idea in this paper is much nicer than R3NN.

---

> > > ### Author Response · Authors · 2018-11-16
> > > **More answers**
> > >
> > > lastToken, lastUse, lastSibling and parent return unique nodes. While you are right that lastUse could be understood to return all locations in which a variable may have been used in an execution (which would indeed require several edges), we mean the lexically last occurrence of the node, which is uniquely determined. This is a simplification of the approach of Allamanis et al. (2018), though we are not stating this explicitly in our submission. We will clarify this in the next revision.
> > >
> > > [The actual source code used to compute is posted anonymously on http://paste.debian.net/hidden/7f6ba717/ now]
> > >
> > >
> > > "inheritedAttr" returns the node corresponding to the inherited attribute of a node, e.g., for node 10 in step 8 of Fig. 2, inheritedAttr(10) would return 0. We will clarify this in the next revision as well.
> > >
> > >
> > > Thank you for your very detailed questions, they do help as us a lot to identify parts of the paper that are insufficiently precise.

---

> ### Author Response · Authors · 2018-11-15
> **Clarification and answers to review questions (Part 2)**
>
> [Second part of reply, as we were over 5000 chars]
>
> > In terms of evaluation, the task seems to be on a different set of
> > expressions than the one explained in the exposition. How many
> > expressions where there in the evaluation programs and how many were
> > chosen to evaluate on and based on what criteria.
>
> This is discussed in the first paragraph of Sect. 5:
> - "all expressions of the fragment that we are considering (i.e.,
>    restricted to numeric, Boolean and string types, or arrays of such
>    values; and not using any user-defined functions)"
> - "343974 samples overall [...] ~100k samples generated from 114 projects
>    into a 'test-only' [...] remaining data we split into
>    training-validation-test sets (60-20-20)"
>
> We are not sure what additional information you are asking for here. Could you please elaborate?
>
> > It seems from the exposition that expressions with field accessed and
> > function calls are not possible to be generated, but then some
> > completions show method calls.
>
> We exclude _user-defined_ functions but allow the built-in functions (and fields) of the considered data types, which primarily include string manipulation/tests ("Substring", "IndexOf", etc.) and generic functions such as "Equals" or the "Length" field of arrays. These built-in function calls are added as new productions in the underlying C# expression grammar.
>
> > In particular, several of the cited prior works solve specific
> > problems like constants that are ignored here.
>
> We do handle constants (i) by generation from a vocabulary and (ii) by copying from context (cf. "Choosing Productions, Variables & Literals" and equation (5)); what other problem do you have in mind here?
>
> > The evaluation is mostly an ablation studies of the proposed approach
> > by removing edges from the final idea. Besides this, the paper also
> > introduces a new dataset for showcasing the technique and does not
> > report sizes and running times, essentially not answering basic
> > questions like what is the trade-off between the different techniques.
>
> We will update the paper to include additional statistics about the experiments (for example, how many epochs were needed to train to convergence, how long an epoch takes on our dataset). Are there any specific statistics that you are interested in besides runtime?
>
> > Comparison to actual prior works on similar tasks is also lacking
> > (some TODO is left in the paper), but there is the claim that existing
> > neural techniques such as seq2seq perform "substantially worse".
>
> A seq2seq baseline achieves 21.8% accuracy (28.1% accuracy in the 5 most probable results returned by beam search) on the test dataset. The perplexity is very high 87.5, primarily driven by uncertainty about generating variables in generated expressions. On the test-only dataset, these are 10.8% accuracy (16.8% @5) and perplexity 130.5.
> We have also updated the paper with the PHOG results.

---

> > ### Comment · AnonReviewer3 · 2018-11-16
> > **Thanks for the updates. Mostly questions about eval:**
> >
> > - choosing expressions:
> >
> > I think part of the question is answered to other reviewers about size of expressions. But for x > y there are 3 expressions that match the description. x, y and x < y. I guess you only take x<y. What is the vocabulary size?
> >
> > - functions and constants:
> >
> > Are functions generated with pickLiteral ? There is no special rule for choosing a built-in function in the generation process, yet these functions are in the dataset. Is there a separate vocabulary for functions and other literals? When copying from the context, do you only include functions or all literals? When picking the dataset, how do you decide if a function is user-defined?
> >
> > Thanks for improving the experiments.

---

> > > ### Author Response · Authors · 2018-11-16
> > > **More answers**
> > >
> > > On choosing expressions:
> > >
> > > We are greedily picking the largest allowed expression from the ASTs that we consider. So for example, from "if ((boolVar || x > y) && UserDefinedFoo(x + y + z - 1))", we select "boolVar || x > y" and "x + y + z - 1" and no other subexpressions.
> > >
> > >
> > > On vocabulary size:
> > > In the graph-expansion setting, there is no classical decoder vocabulary. There is a grammar that we infer from the expressions observed in the training data. This yields rules such as "|Expr| -> ! |Expr|", "|Expr| -> |Expr| + |Expr|", "|Expr| -> |Expr|.Equals(|Expr|", where |Expr| is a non-terminal. That inferred grammar has 222 expressions in total for our dataset, and includes the built-in functions applicable to the datatypes we support.
> > >
> > > Some non-terminals are treated specially, as discussed in the paper. Concretely, |Variable| is expanded using pickVariable from Eq. 4, and |${Type}Literal| is expanded using pickLiteral from Eq. 5. The vocabularies used in pickLiteral have size 50, i.e., we pick from the 50 most common integer/character/string literals observed in the training data.
> > > The copying part of the pickLiteral has access to all tokens in the context that are not language keywords ("for", "public", etc). Anecdotally, we can report that this makes no significant difference -- this masking of keywords/nonterminal nodes in the context was disabled by a bug for some experimental runs without negative effects.
> > >
> > >
> > > On generating functions:
> > > See above - the limited number of functions have dedicated grammar rules, i.e., "|Expr|.Equals(|Expr|)" is treated analogous to generating "|Expr| + |Expr|".
> > >
> > >
> > > On determining "user-definedness" of methods:
> > > As we only support methods, there is a straightforward check. When we observe "var.Method(${args})", we check if "var" is of an allowed type; this implies that Method is implemented in the type and not by the user. If the arguments ${args} are also in our fragment of the language, we include the full expression in the dataset; the inferrence of grammar rules from the observed ASTs then yields a rule |Expr| -> |Expr|.Method(...)".
> > >
> > > This is actually not completely correct, as C# has an extension mechanism by which methods can be added to existing type. However we found this to be seldomly used on the types we are restricting ourselves to, though this leads to a handful of user-defined extension methods occuring in our grammar (e.g., there is "|Expr| -> |Expr|.VirtualPathToDbPath()")

---

> ### Author Response · Authors · 2018-11-19
> **Further questions?**
>
> Thank you again for your valuable and detailed feedback. We will add the content of the additional answers we've given in the comments here to the next revision of the paper. Concretely, we will provide more details on (1) the selection of samples in our dataset as well as how we infer the grammar (this will clarify the issue of method calls as well), (2) the exact meaning of lastUse in Alg. 2, with a note on the relationship to Allamanis et al. 2018 (3) the relation to R3NN. Are there any other open questions you had that we overlooked?
>
> Finally, as many of the points you raised in your initial review have been clarified / resolved, would you consider raising your rating for the updated version of our paper?

---

> > ### Comment · AnonReviewer3 · 2018-11-19
> > **Yes**
> >
> > Looking forward to revisions

---

### Official Review · AnonReviewer2 · 2018-11-02
**Interesting task and dataset.**

**Rating:** 7
**Confidence:** 4

**Review:**

The paper introduces a 'code generation as hole completion' task and associated dataset, ExprGen. The authors proposed a novel extension of AST code generation which uses what they call Neural Attribute Grammars. They show the proposed method does well on this task, compared to ablations of their model (which are similar to previous AST approaches).

The task and dataset are interesting, and the comparison of the proposed method to baselines seems thorough.

*Details to Improve*
The authors have a qualitative evaluation section describing the differences in errors made by various methods. Making this more quantitative by categorizing the errors and computing their frequency would be quite interesting.

---

> ### Author Response · Authors · 2018-11-15
> **Error categorization**
>
> Thank you for your kind review!
>
> > The authors have a qualitative evaluation section describing the
> > differences in errors made by various methods. Making this more
> > quantitative by categorizing the errors and computing their frequency
> > would be quite interesting.
>
> We thought about this as well, but we found it hard to automatically categorize errors beyond considering syntax, type and non-typing semantic errors. We have not reported the numbers for syntax errors in this paper, as all models produce syntactically valid expressions in over 99% of the cases. If you have ideas for metrics that are effectively computable, we are happy to provide additional experimental data.

---

### Official Review · AnonReviewer1 · 2018-11-03
**Novel Model for Programs and Impressive Results**

**Rating:** 7
**Confidence:** 4

**Review:**

In this paper, authors propose a conditional generative model which predicts the missing expression given the surrounding code snippet. Authors represent programs as graphs and use some off-the-shelf encoder to obtain representations for all nodes. Inspired from the attribute grammar, authors augment every node in AST with two new nodes which contain inherited and synthesized information. Based on GGNN, a grammar-driven decoder is further proposed to sequentially generate the AST and the corresponding program. Authors also propose a large dataset which is built from open sourced projects. Experimental results on this dataset show that the proposed method achieves better predictive performance compared to several recent work.

Strength:

1, The problem this paper tries to tackle, i.e., building generative models of code, is very challenging and of great significance.

2, The overall model is a novel and successful attempt to incorporate the structure information of the program into neural networks. I think it will be inspiring for other machine learning based programming applications.

3, The results are very promising and impressive, especially given the large size of the proposed dataset. For example, the top 5 accuracy of predicting correct expression on unseen projects is 57%.

Weakness:

1, I think it would be great to provide more statistics of the proposed dataset, e.g., the average number of tokens, the average size of ASTs.

2, Given the dynamic nature of the graph generation process, I am curious about the efficiency of the proposed method. It would be great to provide some run time information. Also, since recurrent networks are heavily used throughout the model, I wonder how difficult the training process is.

3, It would be great to also compare the log likelihood on the test set.

4, It is unclear from the paper that whether authors use a pre-trained GGNN as encoder or train the encoder end-to-end with the decoder from scratch.

5, It would be great to improve figure 2 as it is not easy to read. Maybe draw another graph to illustrate the temporal evolution of AST?

Overall, I think this paper has made a great progress towards neural modelling of programs and recommend it to be accepted for ICLR.

---

> ### Author Response · Authors · 2018-11-15
> **Answers to review questions**
>
> Thank you for your careful review and kind comments. We hope that we can further improve our submission with your feedback.
>
> > 1, I think it would be great to provide more statistics of the
> > proposed dataset, e.g., the average number of tokens, the average size
> > of ASTs.
>
> As a reminder, there are 344k samples in the dataset overall. There are on average 4.32 (stddev 3.80) tokens per expression to generate [2 tokens: 88k, 3 tokens: 116k, 4 tokens: 38k, 5 tokens: 32k, 6 tokens: 21k, 7 tokens or more: 49k]. The generated trees on average have 3.69 (stddev 3.06) production steps. The dataset is clearly dominated by simple expressions such as “x > y” and “x[y]” (we have filtered out single-variable expressions), but longer expressions are included often as well (cf. Sect. 2 for more details on the selection process). We can also report that we have successfully extended the model to generate whole blocks of statements in a different research project.
>
> > 2, Given the dynamic nature of the graph generation process, I am
> > curious about the efficiency of the proposed method. It would be great
> > to provide some run time information. Also, since recurrent networks
> > are heavily used throughout the model, I wonder how difficult the
> > training process is.
>
> Regarding performance: Training is relatively efficient, as we know the target expansion graph and can thus compute the representations of all nodes in the expansion graph in one go. While that computation is relatively easy to parallelize, its length is the length of the longest path in the target expansion graph. We currently cap this at 50 during training (which excludes only few examples in our dataset). Combined with the computationally relatively expensive GGNN-based encoder, training on a K80 processes around ~25 samples/s, compared to about ~60 samples/s just for the encoder.
>
> As we needed to implement beam search at test time, we have decided to entirely forego batching (and, indeed, GPU usage) at test time, and have essentially implemented Alg. 1 from the paper directly in Python, pruning back the set of beams after each expansion step. For this, “getRepresentation” is a lazy implementation of Eq. (2), computing node representations by message passing “on demand”. We made no efforts to optimize this implementation, instead aiming for simplicity to avoid bugs. An implementation in a dynamic computation graph framework such as TF Eager or PyTorch should be able to significantly outperform our code.
>
> We will report precise runtime statistics in the supplementary material in our next revision. We also plan to release our implementation.
>
> > 3, It would be great to also compare the log likelihood on the test
> > set.
>
> The perplexity shown in Table 1 is directly proportional to the log likelihood in the test set, albeit normalized per token. Could you please elaborate on what you had in mind and how this differs from the results in Table 1?
>
> > 4, It is unclear from the paper that whether authors use a pre-trained
> > GGNN as encoder or train the encoder end-to-end with the decoder from
> > scratch.
>
> The full network is trained end-to-end with no pretraining. We will clarify this in the text.
>
> > 5, It would be great to improve figure 2 as it is not easy to read.
> > Maybe draw another graph to illustrate the temporal evolution of AST?
>
> We’re sorry that this figure is not legible. Given the suggested page limit we wanted to make this explanation as concise as possible. In the new version, we redrew this figure as a sequence of multiple minifigures that show the evolution of the propagation.
> (This has extended the content length beyond the recommended 8 pages, but we agree with you that clarity of this illustration is more important.)

---

> > ### Comment · AnonReviewer1 · 2018-11-19
> > **Thanks for the update**
> >
> > The new figure 2 is indeed much clearer. Thanks!

---

### Author Response · Authors · 2018-11-02
**PHOG experimental results**

The authors of the non-neural PHOG model have now run additional experiments on the dataset used in our paper. Note that their model is a language model and thus only takes code "left of" the hole to fill into account, and that their framework is fairly generic and does not have special modeling of which variables are in scope etc. Hence, the results of their model are only a lower bound of what their model could achieve on this task with suitable extensions; e.g., it would be possible to extend their formalism to also take code after the hole to fill into account.

Bearing these limitations in mind, their results (i.e., their row in Table 1) are as follows:
On the "Test" dataset:
 Acc@1: 34.8%
 Acc@5: 42.9%
On the "Test-only" dataset:
 Acc@1: 28.0%
 Acc@5: 37.3%

---

### Author Response · Authors · 2018-11-15
**Overview of changes in first revision**

We have updated our submission taking some of the reviewer's feedback into account and hope that this improves its clarity. Primarily, we have made the following changes:
- Experiments: We have included results for the PHOG and seq2seq baselines in the paper.
- Notation: We have slightly improved the notation in Alg. 2 and Eq. (2) and added a paragraph giving an overview of used notation in Sect. 3.
- Visualization of tree expansion: We have replaced Fig. 2 by a step-by-step version that should be easier to follow for the reader.

In the next revision, we plan to reflect the remaining feedback and make the following changes:
- Statistics about dataset and (runtime) performance of decoders.
- Experiments with a graph2seq baseline model.

---

> ### Comment · AnonReviewer3 · 2018-11-15
> **Thank you**
>
> Thank you for the update. Indeed, the explanations and the notations is much better now.

---

### Meta-Review · Area_Chair1 · 2018-12-13

**Confidence:** 4
**Recommendation:** Accept (Poster)

**Metareview:**

This paper presents an interesting method for code generation using a graph-based generative approach.  Empirical evaluation shows that the method outperforms relevant baselines (PHOG).

There is consensus among reviewers that the methods are novel and is worth acceptance to ICLR.